# Connectivity Learning in Multi-Branch Networks

## Abstract

While much of the work in the design of convolutional networks over the last five years has revolved around the empirical investigation of the importance of depth, filter sizes, and number of feature channels, recent studies have shown that *branching*, i.e., splitting the computation along parallel but distinct threads and then aggregating their outputs, represents a new promising dimension for significant improvements in performance. To combat the complexity of design choices in multi-branch architectures, prior work has adopted simple strategies, such as a fixed branching factor, the same input being fed to all parallel branches, and an additive combination of the outputs produced by all branches at aggregation points.

In this work we remove these predefined choices and propose an algorithm to learn the connections between branches in the network. Instead of being chosen a priori by the human designer, the multi-branch connectivity is learned simultaneously with the weights of the network by optimizing a single loss function defined with respect to the end task. We demonstrate our approach on the problem of multi-class image classification using four different datasets where it yields consistently higher accuracy compared to the state-of-the-art "ResNeXt" multi-branch network given the same learning capacity.

## 1 Introduction

Deep neural networks have emerged as one of the most prominent models for problems that require the learning of complex functions and that involve large amounts of training data. While deep learning has recently enabled dramatic performance improvements in many application domains, the design of deep architectures is still a challenging and time-consuming endeavor. The difficulty lies in the many architecture choices that impact—often significantly—the performance of the system. In the specific domain of image categorization, which is the focus of this paper, significant research effort has been invested in the empirical study of how depth, filter sizes, number of feature maps, and choice of nonlinearities affect performance (Glorot et al., 2011; Krizhevsky et al., 2012; Sermanet et al., 2013; Maas et al., 2013; Zeiler & Fergus, 2014; Szegedy et al., 2015). Recently, several authors have proposed to simplify the architecture design by defining convolutional neural networks (CNNs) in terms of combinations of basic building blocks. This strategy was arguably first popularized by the VGG networks (Simonyan & Zisserman, 2015) which were built by stacking a series of convolutional layers having identical filter size ($3 \times 3$). The idea of modularized CNN design was made even more explicit in residual networks (ResNets) (He et al., 2016), which are constructed by combining residual blocks of fixed topology. While in ResNets residual blocks are stacked one on top of each other to form very deep networks, the recently introduced ResNeXt models (Xie et al., 2017) have shown that it is also beneficial to arrange these building blocks in parallel to build multi-branch convolutional networks. The modular component of ResNeXt then consists of $C$ parallel branches, corresponding to residual blocks with identical topology but distinct parameters. Network built by stacking these multi-branch components have been shown to lead to better results than single-thread ResNets of the same capacity.

While the principle of modularized design has greatly simplified the challenge of building effective architectures for image analysis, the choice of how to combine and aggregate the computations of these building blocks still rests on the shoulders of the human designer. In order to avoid a combinatorial explosion of options, prior work has relied on simple, uniform rules of aggregation

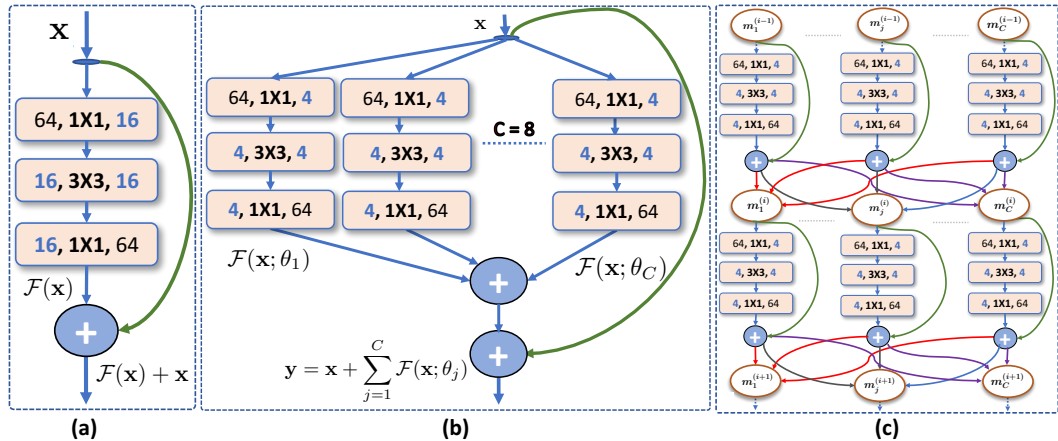

Figure 1: Different types of building blocks for modular network design: (a) a prototypical residual block with bottleneck convolutional layers (He et al., 2016); (b) the multi-branch RexNeXt module consisting of $C$ parallel residual blocks (Xie et al., 2017); (c) our approach replaces the fixed aggregation points of RexNeXt with learnable masks $\mathbf{m}$ defining the input connections for each individual residual block.

and composition. For example, ResNeXt models (Xie et al., 2017) are based on the following set of simplifying assumptions: the branching factor $C$ (also referred to as *cardinality*) is fixed to the same constant in all layers of the network, all branches of a module are fed the same input, and the outputs of parallel branches are aggregated by a simple additive operation that provides the input to the next module. In this paper we remove these predefined choices and propose an algorithm that learns to combine and aggregate building blocks of a neural network. In this new regime, the network connectivity naturally arises as a result of the training optimization rather than being hand-defined by the human designer.

We demonstrate our approach using residual blocks as our modular components, but we take inspiration from ResNeXt by arranging these modules in a multi-branch architecture. Rather than predefining the input connections and aggregation pathways of each branch, we let the algorithm discover the optimal way to combine and connect residual blocks with respect to the end learning objective. This is achieved by means of *masks*, i.e., learned binary parameters that act as "switches" determining the final connectivity in our network. The masks are learned together with the convolutional weights of the network, as part of a joint optimization via backpropagation with respect to a traditional multi-class classification objective. We demonstrate that, given the same budget of residual blocks (and parameters), our learned architecture consistently outperforms the predefined ResNeXt network in all our experiments. An interesting byproduct of our approach is that it can automatically identify residual blocks that are superfluous, i.e., unnecessary or detrimental for the end objective. At the end of the optimization, these unused residual blocks can be pruned away without any impact on the learned hypothesis while yielding substantial savings in number of parameters to store and in test-time computation.

## 2 TECHNICAL APPROACH

### 2.1 MODULAR MULTI-BRANCH ARCHITECTURE

We begin by providing a brief review of residual blocks (He et al., 2016), which represent the modular components of our architecture. We then discuss ResNeXt (Xie et al., 2017), which inspired the multi-branch structure of our networks. Finally, we present our approach to learning the connectivity of multi-branch architectures using binary masks.

**Residual Learning.** The framework of residual learning was introduced by He et al. (He et al., 2016) as a strategy to cope with the challenging optimization of deep models. The approach was

inspired by the observation that deeper neural networks, despite having larger learning capacity than shallower models, often yield higher *training* error, due to the difficulty of optimization posed by increasing depth. Yet, given any arbitrary shallow network, it is trivially possible to reproduce its function using a deeper model, e.g., by copying the shallow network into the top portion of the deep model and by setting the remaining layers to implement identity functions. This simple yet revealing intuition inspired the authors to introduce residual blocks, which learn residual functions with reference to the layer input. Figure 1(a) illustrates an example of these modular components where the 3 layers in the block implement a residual function $\mathcal{F}(\mathbf{x})$. A shortcut connections aggregates the residual block output $\mathcal{F}(\mathbf{x})$ with its input $\mathbf{x}$, thus computing $\mathcal{F}(\mathbf{x}) + \mathbf{x}$, which becomes the input to the next block. The point of this module is that if at any depth in the network the representation $\mathbf{x}$ is already optimal, then $\mathcal{F}(\mathbf{x})$ can be trivially set to be the zero function, which is easier to learn than an identity mapping. In fact, it was shown (He et al., 2016) that reformulating the layers as learning residuals eases optimization and enables the effective training of networks that are substantially deeper than previously possible. Since we are interested in applying our approach to image categorization, in this paper we use convolutional residual blocks using the bottleneck (He et al., 2016) shown in Figure 1(a). The first $1 \times 1$ layer projects the input feature maps onto a lower dimensional embedding, the second applies $3 \times 3$ filters, and the third restores the original feature map dimensionality. As in (He et al., 2016), Batch Normalization (Ioffe & Szegedy, 2015) and ReLU (Krizhevsky et al., 2012) are applied after each layer, and a ReLU is used after each aggregation.

**The multi-branch architecture of ResNeXt.**   Recent work (Xie et al., 2017) has shown that it is beneficial to arrange residual blocks not only along the depth dimension but also to implement parallel multiple threads of computation feeding from the same input layer. The outputs of the parallel residual blocks are then summed up together with the original input and passed on to the next module. The resulting multi-branch module is illustrated in Figure 1(b). More formally, let $\mathcal{F}(\mathbf{x}; \theta_j^{(i)})$ be the transformation implemented by the $j$-th residual block in module $i$-th of the network, where $j = 1, \ldots, C$ and $i = 1, \ldots, L$, with $L$ denoting the total number of modules stacked on top of each other to form the complete network. The hyperparameter $C$ is called the cardinality of the module and defines the number of parallel branches within each module. The hyperparameter $L$ controls the total depth of the network: under the assumption of 3 layers per residual block (as shown in the figure), the total depth of the network is given by $D = 2 + 3L$ (an initial convolutional layer and an output fully-connected layers add 2 layers). Note that in ResNeXt all residual blocks in a module have the same topology ($\mathcal{F}$) but each block has its own parameters ($\theta_j^{(i)}$ denotes the parameters of residual block $j$ in module $i$). Then, the output of the $i$-th module is computed as:

$$\mathbf{y} = \mathbf{x} + \sum_{j=1}^{C} \mathcal{F}(\mathbf{x}; \theta_j^{(i)}) \tag{1}$$

Tensor $\mathbf{y}$ represents the input to the $(i + 1)$-th module. Note that the ResNeXt module effectively implements a *split-transform-merge* strategy that perfoms a projection of the input into separate lower-dimensional embeddings (via bottlenecks), a separate transformation within each embedding, a projection back to the high-dimensional space and a final aggregation via addition. It can be shown that the solutions that can be implemented by such module are a strict subspace of the solutions of a single layer operating on the high-dimensional embedding but at a considerably lower cost in terms of computational complexity and number of parameters. In (Xie et al., 2017) it was experimentally shown that increasing the cardinality $C$ is a more effective way of improving accuracy compared to increasing depth or the number of filters. In other words, given a fixed budget of parameters, ResNeXt multi-branch networks were shown to consistently outperform single-branch ResNets of the same learning capacity.

We note, however, that in an attempt to ease network design, several restrictive limitations were embedded in the architecture of ResNeXt modules: each ResNeXt module implements $C$ parallel feature extractors that operate on the same input; furthermore, the number of active branches is constant at all depth levels of the network. In the next subsection we present an approach that removes these restrictions without adding any significant burden on the process of manual network design (with the exception of a single additional integer hyperparameter for the entire network).

**Our masked multi-branch architecture.**   As in ResNeXt, our proposed architecture consists of a stack of $L$ multi-branch modules, each containing $C$ parallel feature extractors. However, differently from ResNeXt, each branch in a module can take a different input. The input pathway of each

branch is controlled by a binary mask vector that is learned jointly with the weights of the network. Let $\mathbf{m}_j^{(i)} = [m_{j,1}^{(i)}, m_{j,2}^{(i)}, \ldots, m_{j,C}^{(i)}]^\top \in \{0,1\}^C$ be the binary mask vector defining the *active* input connections feeding the $j$-th residual block in module $i$. If $m_{j,k}^{(i)} = 1$, then the activation volume produced by the $k$-th branch in module $(i-1)$ is fed as input to the $j$-th residual block of module $i$. If $m_{j,k}^{(i)} = 0$, then the output from the $k$-th branch in the previous module is ignored by the $j$-th residual block of the current module. Thus, if we denote with $\mathbf{y}_k^{(i-1)}$ the output activation tensor computed by the $k$-th branch in module $(i-1)$, the input $\mathbf{x}_j^{(i)}$ to the $j$-th residual block in module $i$ will be given by the following equation:

$$\mathbf{x}_j^{(i)} = \sum_{k=1}^{C} m_{j,k}^{(i)} \cdot \mathbf{y}_k^{(i-1)} \qquad (2)$$

Then, the output of this block will be obtained through the usual residual computation, i.e., $\mathbf{y}_j^{(i)} = \mathbf{x}_j^{(i)} + \mathcal{F}(\mathbf{x}_j^{(i)}; \theta_j^{(i)})$. We note that under this model we no longer have fixed aggregation nodes summing up *all* outputs computed from a module. Instead, the mask $\mathbf{m}_j^{(i)}$ now determines *selectively* for each block which branches from the previous module will be aggregated and provided as input to the block. Under this scheme, the parallel branches in a module receive different inputs and as such are likely to yield more diverse features.

We point out that depending on the constraints posed over $\mathbf{m}_j^{(i)}$, different interesting models can be realized. For example, by introducing the constraint that $\sum_k m_{j,k}^{(i)} = 1$ for all blocks $j$, then each residual block will receive input from only one branch (since each $m_{j,k}^{(i)}$ must be either 0 or 1). It can be noted that at the other end of the spectrum, if we set $m_{j,k}^{(i)} = 1$ for all blocks $j, k$ in each module $i$, then all connections would be active and we would obtain again the fixed ResNeXt architecture. In our experiments we will demonstrate that the best results are achieved for a middle ground between these two extremes, i.e., by connecting each block to $K$ branches where $K$ is an integer-valued hyperparameter such that $1 < K < C$. We refer to this hyperparameter as the *fan-in* of a block. As discussed in the next section, the mask vector $\mathbf{m}_j^{(i)}$ for each block is learned simultaneously with all the other weights in the network via backpropagation. Finally, we note that it may be possible for a residual block in the network to become unused. This happens when, as a result of the optimization, block $k$ in module $(i-1)$ is such that $m_{jk}^{(i)} = 0$ for all $j = 1, \ldots, C$. In this case, at the end of the optimization, we prune the block in order to reduce the number of parameters to store and to speed up inference (note that this does not affect the function computed by the network). Thus, at any point in the network the total number of active parallel threads can be any number smaller than or equal to $C$. This implies that a variable branching factor is learned adaptively for the different depths in the network.

## 2.2 MASKCONNECT: LEARNING TO CONNECT BRANCHES

We refer to our learning algorithm as MASKCONNECT. It performs joint optimization of a given learning objective $\ell$ with respect to both the weights of the network ($\theta$) as well as the masks ($\mathbf{m}$). Since in this paper we apply our method to the problem of image categorization, we use the traditional multi-class cross-entropy objective for the loss $\ell$. However, our approach can be applied without change to other loss functions as well as to other tasks benefitting from a multi-branch architecture.

In MASKCONNECT the weights have real values, as in traditional networks, while the branch masks have binary values. This renders the optimization more challenging. To learn these binary parameters, we adopt a modified version of backpropagation, inspired by the algorithm proposed by Courbariaux et al. (Courbariaux et al., 2015) to train neural networks with binary weights. During training we store and update a real-valued version $\tilde{\mathbf{m}}_j^{(i)} \in [0,1]^C$ of the branch masks, with entries clipped to lie in the continuous interval from 0 to 1.

In general, the training via backpropagation consists of three steps: 1) forward propagation, 2) backward propagation, and 3) parameters update. At each iteration, we stochastically binarize the real-valued branch masks into binary-valued vectors $\mathbf{m}_j^{(i)} \in \{0,1\}^C$ which are then used for the forward propagation and backward propagation (steps 1 and 2). Instead, during the parameters update (step 3), the method updates the real-valued branch masks $\tilde{\mathbf{m}}_j^{(i)}$. The weights $\theta$ of the convolutional and fully connected layers are optimized using standard backpropagation. We discuss below the details of our mask training procedure, under the constraint that at any time there can be only $K$ active entries in the binary branch mask $\mathbf{m}_j^{(i)}$, where $K$ is a predefined integer hyperparameter with

$1 \leq K \leq C$. In other words, we impose the following constraints:

$$m_{j,k}^{(i)} \in \{0, 1\}, \quad \sum_{k=1}^{C} m_{j,k}^{(i)} = K \quad \forall j \in \{1, \dots, C\} \text{ and } \forall i \in \{1, \dots, L\}.$$

These constraints imply that each residual block receives input from exactly $K$ branches of the previous module.

**Forward Propagation.** During the forward propagation, our algorithm first normalizes the $C$ real-valued branch masks for each block $j$ to sum up to 1, i.e., such that $\sum_{k=1}^{C} \tilde{m}_{j,k}^{(i)} = 1$. This is done so that $\text{Mult}(\tilde{m}_{j,1}^{(i)}, \tilde{m}_{j,2}^{(i)}, \dots, \tilde{m}_{j,C}^{(i)})$ defines a proper multinomial distribution over the $C$ branch connections feeding into block $j$. Then, the binary branch mask $\mathbf{m}_j^{(i)}$ is stochastically generated by drawing $K$ *distinct* samples $a_1, a_2, \dots, a_K \in \{1, \dots, C\}$ from the multinomial distribution over the branch connections. Finally, the entries corresponding to the $K$ samples are activated in the binary branch mask vector, i.e., $m_{j,a_k}^{(i)} \leftarrow 1$, for $k = 1, \dots, K$. The input activation volume to the residual block $j$ is then computed according to Eq. 2 from the sampled binary branch masks. We note that the sampling from the Multinomial distribution ensures that the connections with largest $\tilde{m}_{j,k}^{(i)}$ values will be more likely to be chosen, while at the same time the stochasticity of this process allows different connectivities to be explored, particularly during early stages of the learning when the real-valued masks have still fairly uniform values.

**Backward Propagation.** In the backward propagation step, the gradient $\partial \ell / \partial y_k^{(i-1)}$ with respect to each branch output is obtained via back-propagation from $\partial \ell / \partial x_j^{(i)}$ and the binary masks $m_{j,k}^{(i)}$.

**Mask Update.** In the parameter update step our algorithm computes the gradient with respect to the binary branch masks for each branch. Then, using these computed gradients and the given learning rate, it updates the real-valued branch masks via gradient descent. At this time we clip the updated real-valued branch masks to constrain them to remain within the valid interval $[0, 1]$. The same clipping strategy was adopted for the binary weights in the work of Courbariaux et al. (2015).

As discussed in the supplementary material, after joint training over $\theta$ and $\mathbf{m}$, we have found beneficial to fine-tune the weights $\theta$ of the network with fixed binary masks (connectivity), by setting as active connections for each block $j$ in module $i$ those corresponding to the $K$ largest values in $\tilde{\mathbf{m}}_j^{(i)}$. Pseudocode for our training procedure is given in the supplementary material.

## 3 EXPERIMENTS

We tested our approach on the task of image categorization using several benchmarks: CIFAR-10 (Krizhesvsky, 2009), CIFAR-100 (Krizhesvsky, 2009), Mini-ImageNet (Vinyals et al., 2016), as well as the full ImageNet (Deng et al., 2009). In this section we discuss results achieved on CIFAR-100 and ImageNet (Deng et al., 2009), while the results for CIFAR-10 (Krizhesvsky, 2009) and Mini-ImageNet (Vinyals et al., 2016) can be found in the Appendix.

### 3.1 CIFAR-100

CIFAR-100 is a dataset of color images of size 32x32. It consists of 50,000 training images and 10,000 test images. Each image in CIFAR-100 is categorized into one of 100 possible classes.

**Effect of fan-in** ($K$). We start by studying the effect of the fan-in hyperparameter ($K$) on the performance of models built and trained using our proposed approach. The fan-in defines the number of *active* branches feeding each residual block. For this experiment we use a model obtained by stacking $L = 6$ multi-branch residual modules, each having cardinality $C = 8$ (number of branches in each module). We use residual blocks consisting of 3 convolutional layers with a bottleneck implementing dimensionality reduction on the number of feature channels, as shown in Figure 1. The bottleneck for this experiment was set to $w = 4$. Since each residual block consists of 3 layers, the total depth of the network in terms of learnable layers is $D = 2 + 3L = 20$.

We trained and tested this architecture using different fan-in values: $K = 1, \dots, 8$. Note that varying $K$ does not affect the number of parameters. Thus, all these models have the same learning capacity.

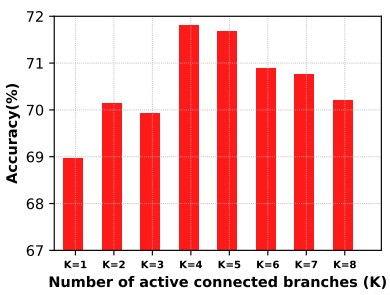

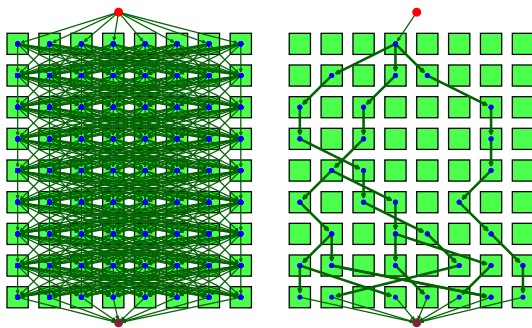

Figure 2: Varying the fan-in ($K$) of our model, i.e., the number of active branches provided as input to each residual block. The plot reports accuracy achieved on CIFAR-100 using a network stack of $L = 6$ ResNeXt modules having cardinality $C = 8$ and bottleneck width $w = 4$. All models have the same number of parameters (0.28M). The best accuracy is obtained for $K = 4$.

Figure 3: A visualization of the fixed branch connectivity of ResNext (left) versus the connectivity learned by our method (right) using ($K = 1$). Each green square is a residual block, each row of $C = 8$ square is a multi-branch module. The network consists of a stack of $L = 9$ modules. Arrows indicate pathways connecting residual blocks of adjacent modules. In each net, the top red circle is a convolutional layer, the bottom circle is the final fully-connected layer. It can be noticed that MASKCONNECT learns sparse connections. The squares without in/out edges are those deemed superfluous by our algorithm and can be pruned at the end of learning. This gives rise to a branching factor that varies along the depth of the net.

The results are shown in Figure 2. We can see that the best accuracy is achieved by connecting each residual block to $K = 4$ branches out of the total $C = 8$ in each module. Using a very low or very high fan-in yields lower accuracy. Note that when setting $K = C$, there is no need to learn the masks. In this case each mask is simply replaced by an element-wise addition of the outputs from all the branches. This renders the model equivalent to ResNeXt (Xie et al., 2017), which has fixed connectivity. Based on the results of Figure 2, in all our experiments below we use $K = 4$, since it gives the best accuracy, but also $K = 1$, since it gives high sparsity which, as we will see shortly, implies savings in number of parameters.

**Varying the architectures.** In Table 1 we show the classification accuracy achieved with different architectures (the details of each architecture are listed in the Appendix). For each architecture we report results obtained using MASKCONNECT with fan-in $K = 1$ and $K = 4$. We also include the accuracy achieved with full (as opposed to learned) connectivity, which corresponds to ResNeXt. These results show that learning the connectivity produces consistently higher accuracy than using fixed connectivity, with accuracy gains of up 2.2% compared to the state-of-the-art ResNeXt model.

We note that these improvements in accuracy come at little computational training cost: the average training time overhead for learning masks and weights is about $39\%$ using our unoptimized implementation compared to learning only the weights given a fixed connectivity. Additionally, for each architecture we include models trained using sparse random connectivity (Fixed-Random). For these models, each mask is set to have $K = 4$ randomly-chosen active connections, and the connectivity is kept fixed during learning of the parameters. We can notice that the accuracy of these nets is considerably lower compared to our models, despite having the same connectivity density ($K = 4$). This shows that the improvements of our approach over ResNeXt are not due to sparser connectivity but they are rather due to *learned* connectivity.

**Parameter savings.** Our proposed approach provides the benefit of automatically identifying during training residual blocks that are unnecessary. At the end of the training, the unused residual blocks can be pruned away. This yields savings in the number of parameters to store and in test-time computation. In Table 1, columns *Train* and *Test* under *Params* show the original number of parameters (used during training) and the number of parameters after pruning (used at test-time). Note that for the biggest architecture, our approach using $K = 1$ yields a parameter saving of 40% compared to ResNeXt with full connectivity (20.5M vs 34.4M), while achieving the same accuracy. Thus, in summary, using fan-in $K = 4$ gives models that have the same number of parameters as ResNeXt but they yield higher accuracy; using fan-in $K = 1$ gives a significant saving in number of parameters and accuracy on par with ResNeXt.

**Model with real-valued masks.** We have also attempted to learn our models using real-valued masks by computing tensors in the forward and backward propagation with respect to masks $\tilde{\mathbf{m}}_j^{(i)} \in [0,1]^C$ rather than the binary vectors $\mathbf{m}_j^{(i)} \in \{0,1\}^C$. However, we found this variant to

Table 1: CIFAR-100 accuracies (single crop) achieved by different architectures trained using the predefined full connectivity of ResNeXt (Fixed-Full) versus the connectivity learned by our algorithm (Learned). We also include models trained using random, fixed connectivity (Fixed-Random) defined by setting $K = 4$ random active connections per branch. Each model was trained 4 times, using different random initializations. For each model we report the best test performance as well as the mean test performance computed from the 4 runs. For our method, we report performance using $K = 1$ as well as $K = 4$. We also list the number of parameters used during training (Params-Train) and the number of parameters obtained after pruning the unused blocks (Params-Test). Our learned connectivity using $K = 4$ produces accuracy gains of up 2.2% compared to the strong ResNeXt model, while using $K = 1$ yields results equivalent to ResNeXt but it induces a significant reduction in number of parameters at test time (a saving of 40% for model {29,64,8}).

| Architecture | Connectivity | Params | | Accuracy (%) |
|---|---|---|---|---|
| {Depth ($D$), Bottleneck width ($w$), Cardinality ($C$)} | | *Train* | *Test* | *Top-1* best (mean±std) |
| {29,8,8} | Fixed-Full, K=8 (Xie et al., 2017) | 0.86M | 0.86M | 73.52 (73.37±0.13) |
| | **Learned**, K=1 | 0.86M | 0.65M | **73.91** (73.76±0.14) |
| | **Learned**, K=4 | 0.86M | 0.81M | **75.89** (75.77±0.12) |
| | Fixed-Random, K=4 | 0.86M | 0.85M | 72.85 (72.66±0.24) |
| {29,64,8} | Fixed-Full, K=8 (Xie et al., 2017) | 34.4M | 34.4M | 82.23 (82.12±0.12) |
| | **Learned**, K=1 | 34.4M | 20.5M | **82.31** (82.15±0.15) |
| | **Learned**, K=4 | 34.4M | 32.1M | **84.05** (83.94±0.11) |
| | Fixed-Random, K=4 | 34.4M | 34.3M | 81.96 (81.73±0.20) |

yield consistently lower results compared to our models using binary masks. For example, for model $\{D = 29, w = 8, C = 8\}$ the best accuracy achieved with real-valued masks is 1.93% worse compared to that obtained with binary masks. In particular we observed that for this variant, the real-valued masks change little over training even when using large learning rates. Conversely, performing the forward and backward propagation using stochastically-sampled binary masks yields a larger exploration of connectivities and results in bigger changes of the auxiliary real-valued masks leading to better connectivity learning.

**Visualization of the learned connectivity.** Figure 3 provides an illustration of the connectivity learned by MASKCONNECT for $K = 1$ versus the fixed connectivity of ResNeXt for model $\{D = 29, w = 8, C = 8\}$. While ResNeXt feeds the same input to all blocks of a module, our algorithm learns different input pathways for each block and yields a branching factor that varies along depth.

## 3.2 IMAGENET

Finally, we evaluate our approach on the large-scale ImageNet 2012 dataset (Deng et al., 2009), which includes images of 1000 classes. We train our approach on the training set (1.28M images) and evaluate it on the validation set (50K images). In Table 2, we report the Top-1 and Top-5 accuracies for three different architectures. For these experiments we set $K = C/2$. We can observe that for all three architectures, our learned connectivity yields an improvement in accuracy over the fixed connectivity of ResNeXt (Xie et al., 2017).

## 3.3 CIFAR-10 & MINI-IMAGENET

We invite the reader to review results achieved on CIFAR-10 & Mini-ImageNet in the Appendix. Also on these datasets our algorithm consistently outperforms the ResNeXt models based on fixed connectivity, with accuracy gains of up to 3.8%.

## 4 RELATED WORK

Despite their wide adoption, deep networks often require laborious model search in order to yield good results. As a result, significant research effort has been devoted to the design of algorithms for automatic model selection. However, most of this prior work falls within the genre of hyper-

Table 2: ImageNet accuracies (single crop) achieved by different architectures using the predefined connectivity of ResNeXt (Fixed-Full) versus the connectivity learned by our algorithm (Learned).

| Architecture | Connectivity | Accuracy | |
|---|---|---|---|
| {Depth ($D$), Bottleneck width ($w$), Cardinality ($C$)} | | *Top-1* | *Top-5* |
| {50,4,32} | Fixed-Full, K=32 (Xie et al., 2017) | 77.8 | 93.3 |
| | **Learned**, K=16 | **79.1** | **94.1** |
| {101,4,32} | Fixed-Full, K=32 (Xie et al., 2017) | 78.8 | 94.1 |
| | **Learned**, K=16 | **79.5** | **94.5** |
| {101,4,64} | Fixed-Full, K=64 (Xie et al., 2017) | 79.6 | 94.7 |
| | **Learned**, K=32 | **79.8** | **94.8** |

parameter optimization (Bergstra & Bengio, 2012; Snoek et al., 2012; 2015) rather than architecture or connectivity learning. Evolutionary search has been proposed as an interesting framework to learn both the structure as well as the connections in a neural network (Wierstra et al., 2005; Floreano et al., 2008; Real et al., 2017). Architecture search has also been recently formulated as a reinforcement learning problem with impressive results (Zoph & Le, 2017). Unlike these approaches, our method is limited to learning the connectivity within a predefined architecture but it does so efficiently by gradient descent optimization of the learning objective as opposed to more costly procedures such as evolutionary search or reinforcement learning. Several authors have proposed learning connectivity by pruning unimportant weights from the network (LeCun et al., 1989; Han et al., 2015a;b; Guo et al., 2016; Han et al., 2016). However, these prior methods operate in stages where initially the network with full connectivity is learned and then connections are greedily removed according to an importance criterion. In PathNet (Fernando et al., 2017), the connectivity within a given architecture was searched via evolution. Compare to these prior approaches, our work provides the advantage of learning the connectivity by direct global optimization of the loss function of the problem at hand rather than by greedy optimization of a proxy criterion or by evolution. Our technical approach shares similarities with the "Shake-Shake" regularization recently introduced in unpublished work (Gastaldi, 2017). This procedure was demonstrated on two-branch ResNeXt models and consists in randomly scaling tensors produced by parallel branches during each training iteration while at test time the network uses uniform weighting of tensors. Conversely, our algorithm *learns* an optimal binary scaling of the parallel tensors with respect to the training objective and uses the resulting network with sparse connectivity at test time. Our work is also related to approaches that learn a hierarchical structure in the last one or two layers of a network in order to obtain distinct features for different categories (Murdock et al., 2016; Ahmed & Torresani, 2017). Differently from these methods, our algorithm learns efficiently connections at all depths in the network, thus optimizing over a much larger family of connectivity models. While our algorithm is limited to optimizing the connectivity structure within a predefined architecture, Adams et al. (Adams et al., 2010) proposed a nonparametric Bayesian approach that searches over an infinite network using MCMC. Saxena and Verbeek (Saxena & Verbeek, 2016) introduced convolutional neural fabric which are learnable 3D trellises that locally connect response maps at different layers of a CNN. Similarly to our work, they enable optimization over an exponentially large family of connectivities, albeit different from those considered here.

## 5 CONCLUSIONS

In this paper we introduced an algorithm to learn the connectivity of deep multi-branch networks. The problem is formulated as a single joint optimization over the weights and the branch connections of the model. We tested our approach on challenging image categorization benchmarks where it led to significant accuracy improvements over the state-of-the-art ResNeXt model. An added benefit of our approach is that it can automatically identify superfluous blocks, which can be pruned without impact on accuracy for more efficient testing and for reducing the number of parameters to store.

While our experiments were focused on a particular multi-branch architecture (ResNeXt) and a specific form of building block (residual block), we expect the benefits of our approach to extend to other modules and network structures. For example, it could be applied to learn the connectivity of skip-connections in DenseNets (Huang et al., 2017), which are currently based on predefined

connectivity rules. In this paper, our masks perform non-parametric additive aggregation of the branch outputs. It would be interesting to experiment with learnable (parametric) aggregations of the outputs from the individual branches. Our approach is limited to learning connectivity within a given, fixed architecture. Future work will explore the use of learnable masks for architecture discovery.

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

## A PSEUDOCODE OF THE ALGORITHM

---

**Algorithm 1** MASKCONNECT training algorithm.

---

**Input:** a minibatch of labeled examples $(x^i, y^i)$, $C$: cardinality (number of branches), $K$: fan-in (number of active branch connections), $\eta$: learning rate, $\ell$: the loss over the minibatch, $\tilde{\mathbf{m}}_j^{(i)} \in [0,1]^C$: real-valued branch masks for block $j$ in module $i$ from previous training iteration.

**Output:** updated $\tilde{\mathbf{m}}_j^{(i)}$

**1. Forward Propagation:**

Normalize the real-valued mask to sum up to 1: $\tilde{m}_{j,k}^{(i)} \leftarrow \frac{\tilde{m}_{j,k}^{(i)}}{\sum_{k'=1}^{C} \tilde{m}_{j,k'}^{(i)}}$, for $j = 1, \ldots, C$

Reset binary mask: $\mathbf{m}_j^{(i)} \leftarrow \mathbf{0}$

Draw $K$ *distinct* samples from multinomial mask distribution:

$a_1, a_2, \ldots, a_K \leftarrow \text{Mult}(\tilde{m}_{j,1}^{(i)}, \tilde{m}_{j,2}^{(i)}, \ldots, \tilde{m}_{j,C}^{(i)})$

Set active binary mask based on drawn samples:

$m_{j,a_k}^{(i)} \leftarrow 1$ for $k = 1, \ldots, K$

Compute output $\mathbf{x}_j^{(i)}$ of the mask, given branch activations $\mathbf{y}_k^{(i-1)}$: $\mathbf{x}_j^{(i)} \leftarrow \sum_{k=1}^{C} m_{j,k}^{(i)} \cdot \mathbf{y}_k^{(i-1)}$

**2. Backward Propagation:**

Compute $\frac{\partial \ell}{\partial \mathbf{x}_j^{(i)}}$ from $\frac{\partial \ell}{\partial \mathbf{y}_j^{(i)}}$

Compute $\frac{\partial \ell}{\partial \mathbf{y}_k^{(i-1)}}$ from $\frac{\partial \ell}{\partial \mathbf{x}_j^{(i)}}$ and $m_{j,k}^{(i)}$

**3. Parameter Update:**

Compute $\frac{\partial \ell}{\partial m_{j,k}^{(i)}}$ given $\frac{\partial \ell}{\partial \mathbf{x}_j^{(i)}}$ and $\mathbf{y}_k^{(i-1)}$

$\tilde{m}_{j,k}^{(i)} \leftarrow \text{clip}(\tilde{m}_{j,k}^{(i)} - \eta \cdot \frac{\partial \ell}{\partial m_{j,k}^{(i)}})$

---

## B EXPERIMENTS ON CIFAR-10

The CIFAR-10 dataset consists of color images of size 32x32. The training set contains 50,000 images, the testing set 10,000 images. Each image in CIFAR-10 is categorized into one of 10 possible classes. In Table 3, we report the performance of different models trained on CIFAR-10. From these results we can observe that our models using learned connectivity achieve consistently better performance over the equivalent models trained with the fixed connectivity (Xie et al., 2017).

Table 3: CIFAR-10 accuracies (single crop) achieved by different multi-branch architectures trained using the predefined connectivity of ResNeXt (Fixed-Full) versus the connectivity learned by our algorithm (Learned). Each model was trained 4 times, using different random initializations. For each model we report the best test performance as well as the mean test performance computed from the 4 runs.

| Architecture | Connectivity | Accuracy (%) |
|---|---|---|
| {Depth ($D$), Bottleneck width ($w$), Cardinality ($C$)} | | *Top-1* 
 best (mean±std) |
| {20,4,8} | Fixed-Full K=8 (Xie et al., 2017) 
 **Learned** K=4 | 91.39 (91.13±0.11) 
 **92.85** (92.76±0.10) |
| {29,4,8} | Fixed-Full K=8 (Xie et al., 2017) 
 **Learned** K=4 | 92.77 (92.65±0.09) 
 **93.88** (93.76±0.12) |
| {29,8,8} | Fixed-Full K=8 (Xie et al., 2017) 
 **Learned** K=4 | 93.26 (93.14±0.11) 
 **95.11** (94.96±0.12) |
| {29,64,8} | Fixed-Full K=8 (Xie et al., 2017) 
 **Learned** K=4 | 96.35 (96.23±0.12) 
 **96.83** (96.73±0.11) |

Table 4: Mini-ImageNet accuracies achieved by different multi-branch networks trained using the predefined full connectivity of ResNeXt (Fixed-Full) versus the connectivity learned by our algorithm (Learned). Additionally, we include models trained using random fixed connectivity (Fixed-Random) for $K = 4$. For each model we report the best and the mean test performance computed from 4 different training runs. Our method for joint learning of weights and connectivity yields a gain of over 3% in Top-1 accuracy over ResNeXt, which uses the same architectures but a fixed branch connectivity.

| Architecture | Connectivity | Accuracy |
| --- | --- | --- |
| $\{$Depth $(D)$, Bottleneck width $(w)$, Cardinality $(C)\}$ | | *Top-1* 
 best (mean±std) |
| | Fixed-Full K=8 (Xie et al., 2017) | 62.12 (61.86±0.15) |
| $\{20,4,8\}$ | **Learned** K=4 | **66.09** (65.94±0.16) |
| | Fixed-Random K=4 | 62.42 (61.81±0.32) |
| | Fixed-Full K=8 (Xie et al., 2017) | 68.11 (67.89±0.19) |
| $\{29,8,8\}$ | **Learned** K=4 | **71.36** (71.18±0.19) |
| | Fixed-Random K=4 | 67.97 (67.53±0.20) |

## C    EXPERIMENTS ON MINI-IMAGENET

Mini-ImageNet is a subset of the full ImageNet (Deng et al., 2009) dataset. It was used in (Vinyals et al., 2016; Ravi & Larochelle, 2017). It is created by randomly selecting 100 classes from the full ImageNet (Deng et al., 2009). For each class, 600 images are randomly selected. We use 500 examples per class for training, and the other 100 examples per class for testing. The selected images are resized to size 84x84 pixels as in (Vinyals et al., 2016; Ravi & Larochelle, 2017). The advantage of this dataset is that it poses the recognition challenges typical of the ImageNet photos but at the same time it does not need require the powerful resources needed to train on the full ImageNet dataset. This allows to include the additional baselines involving random fixed connectivity (Fixed-Random).

We report the performance of different models trained on Mini-ImageNet in Table 4. From these results, we see that our models using learned connectivity with fan-in $K$=4 yield a nice accuracy gain over the same models trained with the fixed full connectivity of ResNeXt (Xie et al., 2017). The absolute improvement (in Top-1 accuracy) is 3.87% for the 20-layer network and 3.17% for the 29-layer network. We can notice that the accuracy of the models with fixed random connectivity (Fixed-Random) is considerably lower compared to our nets with learned connectivity, despite having the same connectivity density ($K = 4$). This shows that the improvement of our approach over ResNeXt is not due to sparser connectivity but it is rather due to *learned* connectivity.

## D    VISUALIZATIONS OF LEARNED CONNECTIVITY

The plot in Figure 4 shows how the number of active branches varies as a function of the module depth for model $\{D = 29, w = 4, C = 8\}$ trained on CIFAR-100. For $K = 1$, we can observe that the number of active branches tends to be larger for deep modules (closer to the output layer) compared to early modules (closer to the input). We observed this phenomenon consistently for all architectures. This suggests that having many parallel threads of computation is particularly important in deep layers of the network. Conversely, the setting $K = 4$ tends to produce a fairly uniform number of active branches across the modules and the number is quite close to the maximum value $C$. For this reason, there is little saving in terms of number of parameters when using $K = 4$, as there are rarely unused blocks.

The plot in Figure 5 shows the number of active branches as a function of module depth for model $\{D = 50, w = 4, C = 32\}$ trained on ImageNet, using $K = 16$.

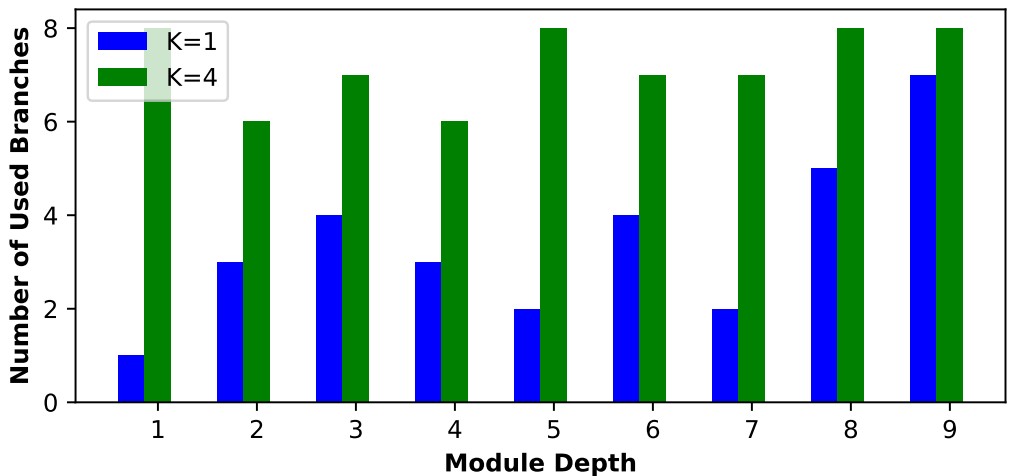

Figure 4: Number of active branches as a function of module depth for model $\{D = 29, w = 4, C = 8\}$ trained on CIFAR-100. We report how the number of active branches varies for model trained with fan-in $K = 1$ as well as for the net trained with $K = 4$. The setting $K = 1$ tends to leave many blocks unused, especially in the early modules of the network.

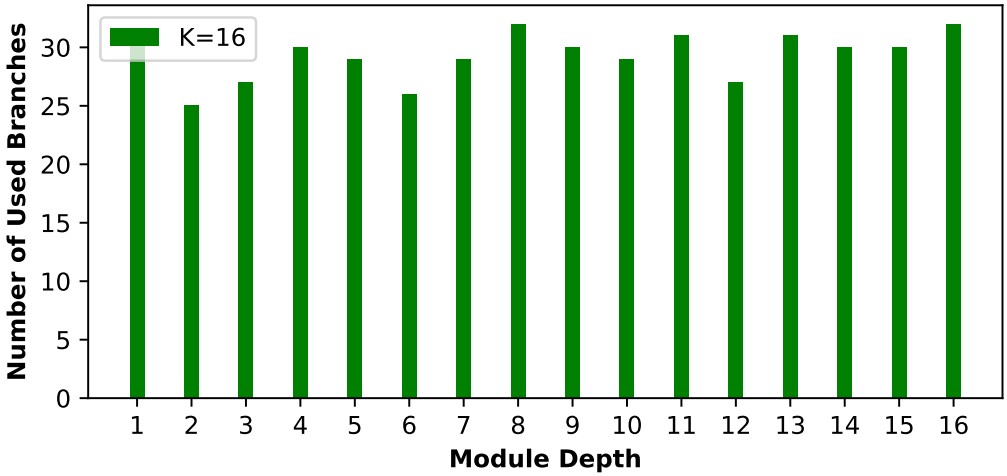

Figure 5: Number of active branches as a function of module depth for model $\{D = 50, w = 4, C = 32\}$ trained on ImageNet, using fan-in $K = 16$.

Table 5: Specifications of the architectures used in our experiments on the CIFAR-10 and CIFAR-100 datasets. The architectures differ in terms of depth ($D$), bottleneck width ($w$), and cardinality ($C$). Inside the brackets we specify the residual block used in each multi-branch module by listing the number of input channels, the size of the convolutional filters, as well as the number of filters (number of output channels). To the right of each bracket we list the cardinality (i.e., the number of parallel branches in the module). $\times 2$ means that the same multi-branch module is stacked twice. The first layer for all models is a convolutional layer with 16 filters of size $3 \times 3$. The last layer performs global average pooling followed by a softmax.

| {$D$=20, $w$=4, $C$=8} | {$D$=29, $w$=4, $C$=8} | {$D$=29, $w$=8, $C$=8} | {$D$=29, $w$=64, $C$=8} |
|---|---|---|---|
| 3, 3×3, 16 | 3, 3×3, 16 | 3, 3×3, 16 | 3, 3×3, 64 |
| $\begin{bmatrix} 16, 1\times1, 4 \\ 4, 3\times3, 4 \\ 4, 1\times1, 64 \end{bmatrix}$ ($C$=8) | $\begin{bmatrix} 16, 1\times1, 4 \\ 4, 3\times3, 4 \\ 4, 1\times1, 64 \end{bmatrix}$ ($C$=8) | $\begin{bmatrix} 16, 1\times1, 8 \\ 8, 3\times3, 8 \\ 8, 1\times1, 64 \end{bmatrix}$ ($C$=8) | $\begin{bmatrix} 64, 1\times1, 64 \\ 64, 3\times3, 64 \\ 64, 1\times1, 256 \end{bmatrix}$ ($C$=8) |
| $\begin{bmatrix} 64, 1\times1, 4 \\ 4, 3\times3, 4 \\ 4, 1\times1, 64 \end{bmatrix}$ ($C$=8) | $\begin{bmatrix} 64, 1\times1, 4 \\ 4, 3\times3, 4 \\ 4, 1\times1, 64 \end{bmatrix}$ ($C$=8), ×2 | $\begin{bmatrix} 64, 1\times1, 8 \\ 8, 3\times3, 8 \\ 8, 1\times1, 64 \end{bmatrix}$ ($C$=8), ×2 | $\begin{bmatrix} 256, 1\times1, 64 \\ 64, 3\times3, 64 \\ 64, 1\times1, 256 \end{bmatrix}$ ($C$=8), ×2 |
| $\begin{bmatrix} 64, 1\times1, 8 \\ 8, 3\times3, 8 \\ 8, 1\times1, 128 \end{bmatrix}$ ($C$=8) | $\begin{bmatrix} 64, 1\times1, 8 \\ 8, 3\times3, 8 \\ 8, 1\times1, 128 \end{bmatrix}$ ($C$=8) | $\begin{bmatrix} 64, 1\times1, 16 \\ 16, 3\times3, 16 \\ 16, 1\times1, 128 \end{bmatrix}$ ($C$=8) | $\begin{bmatrix} 256, 1\times1, 128 \\ 128, 3\times3, 128 \\ 128, 1\times1, 512 \end{bmatrix}$ ($C$=8) |
| $\begin{bmatrix} 128, 1\times1, 8 \\ 8, 3\times3, 8 \\ 8, 1\times1, 128 \end{bmatrix}$ ($C$=8) | $\begin{bmatrix} 128, 1\times1, 8 \\ 8, 3\times3, 8 \\ 8, 1\times1, 128 \end{bmatrix}$ ($C$=8), ×2 | $\begin{bmatrix} 128, 1\times1, 16 \\ 16, 3\times3, 16 \\ 16, 1\times1, 128 \end{bmatrix}$ ($C$=8), ×2 | $\begin{bmatrix} 512, 1\times1, 128 \\ 128, 3\times3, 128 \\ 128, 1\times1, 512 \end{bmatrix}$ ($C$=8), ×2 |
| $\begin{bmatrix} 128, 1\times1, 16 \\ 16, 3\times3, 16 \\ 16, 1\times1, 256 \end{bmatrix}$ ($C$=8) | $\begin{bmatrix} 128, 1\times1, 16 \\ 16, 3\times3, 16 \\ 16, 1\times1, 256 \end{bmatrix}$ ($C$=8) | $\begin{bmatrix} 128, 1\times1, 32 \\ 32, 3\times3, 32 \\ 32, 1\times1, 256 \end{bmatrix}$ ($C$=8) | $\begin{bmatrix} 512, 1\times1, 256 \\ 256, 3\times3, 256 \\ 256, 1\times1, 1024 \end{bmatrix}$ ($C$=8) |
| $\begin{bmatrix} 256, 1\times1, 16 \\ 16, 3\times3, 16 \\ 16, 1\times1, 256 \end{bmatrix}$ ($C$=8) | $\begin{bmatrix} 256, 1\times1, 16 \\ 16, 3\times3, 16 \\ 16, 1\times1, 256 \end{bmatrix}$ ($C$=8), ×2 | $\begin{bmatrix} 256, 1\times1, 32 \\ 32, 3\times3, 32 \\ 32, 1\times1, 256 \end{bmatrix}$ ($C$=8), ×2 | $\begin{bmatrix} 1024, 1\times1, 256 \\ 256, 3\times3, 256 \\ 256, 1\times1, 1024 \end{bmatrix}$ ($C$=8), ×2 |
| Average Pool
100 fc, softmax | Average Pool
100 fc, softmax | Average Pool
100 fc, softmax | Average Pool
100 fc, softmax |

# E   IMPLEMENTATION DETAILS

## E.1   ARCHITECTURES AND SETTINGS FOR EXPERIMENTS ON CIFAR-100 AND CIFAR-10

The specifications of the architectures used in all our experiments on CIFAR-10 and CIFAR-100 are given in Table 5.

Several of these architectures are those presented in the original ResNeXt paper (Xie et al., 2017) and are trained using the same setup, including the data augmentation strategy.Four pixels are padded on each side of the input image, and a 32x32 crop is randomly sampled from the padded image or its horizontal flip, with per-pixel mean subtracted (Krizhevsky et al., 2012). For testing, we use the original 32x32 image. The stacks have output feature map of size 32, 16, and 8 respectively. The models are trained on 8 GPUs with a mini-batch size of 128 (16 per GPU), with a weight decay of 0.0005 and momentum of 0.9. We adopt *four* incremental training phases with a total of 320 epochs. In *phase 1* we train the model for 120 epochs with a learning rate of 0.1 for the convolutional and fully-connected layers, and a learning rate of 0.2 for the masks. In *phase 2* we freeze the connectivity by setting as active connections for each block those corresponding to its top-$K$ values in the masks. With these fixed learned connectivity, we finetune the model from *phase 1* for 100 epochs with a learning rate of 0.1 for the weights. Then, in *phase 3* we finetune the weights of the model from *phase 2* for 50 epochs with a learning rate of 0.01 using again the fixed learned connectivity from phase 1. Finally, in *phase 4* we finetune the weights of the model from *phase 3* for 50 epochs with a learning rate of 0.001.

## E.2   ARCHITECTURES AND SETTINGS FOR EXPERIMENTS ON IMAGENET

The architectures for our ImageNet experiments are those specified in the original ResNeXt paper (Xie et al., 2017).

Table 6: Mini-ImageNet architectures with varying depth ($D$), and bottleneck width ($w$). Inside the brackets we specify the residual block used in each multi-branch module by listing the number of input channels, the size of the convolutional filters, as well as the number of filters (number of output channels). To the right of each bracket we list the cardinality ($C$) (i.e., the number of parallel branches in the module). $\times 2$ means that the same multi-branch module is stacked twice.

| $\{D{=}20, w{=}4, C{=}8\}$ | $\{D{=}29, w{=}8, C{=}8\}$ |
|---|---|
| 3, 3×3, 16 | 3, 3×3, 16 |
| Max Pool, 3×3, stride=2 | Max Pool, 3×3, stride=2 |
| $\begin{bmatrix} 16, 1{\times}1, 4 \\ 4, 3{\times}3, 4 \\ 4, 1{\times}1, 64 \end{bmatrix}$ (C=8) | $\begin{bmatrix} 16, 1{\times}1, 8 \\ 8, 3{\times}3, 8 \\ 8, 1{\times}1, 64 \end{bmatrix}$ (C=8) |
| $\begin{bmatrix} 64, 1{\times}1, 4 \\ 4, 3{\times}3, 4 \\ 4, 1{\times}1, 64 \end{bmatrix}$ (C=8) | $\begin{bmatrix} 64, 1{\times}1, 8 \\ 8, 3{\times}3, 8 \\ 8, 1{\times}1, 64 \end{bmatrix}$ (C=8), ×2 |
| $\begin{bmatrix} 64, 1{\times}1, 8 \\ 8, 3{\times}3, 8 \\ 8, 1{\times}1, 128 \end{bmatrix}$ (C=8) | $\begin{bmatrix} 64, 1{\times}1, 16 \\ 16, 3{\times}3, 16 \\ 16, 1{\times}1, 128 \end{bmatrix}$ (C=8) |
| $\begin{bmatrix} 128, 1{\times}1, 8 \\ 8, 3{\times}3, 8 \\ 8, 1{\times}1, 128 \end{bmatrix}$ (C=8) | $\begin{bmatrix} 128, 1{\times}1, 16 \\ 16, 3{\times}3, 16 \\ 16, 1{\times}1, 128 \end{bmatrix}$ (C=8), ×2 |
| $\begin{bmatrix} 128, 1{\times}1, 16 \\ 16, 3{\times}3, 16 \\ 16, 1{\times}1, 256 \end{bmatrix}$ (C=8) | $\begin{bmatrix} 128, 1{\times}1, 32 \\ 32, 3{\times}3, 32 \\ 32, 1{\times}1, 256 \end{bmatrix}$ (C=8) |
| $\begin{bmatrix} 256, 1{\times}1, 16 \\ 16, 3{\times}3, 16 \\ 16, 1{\times}1, 256 \end{bmatrix}$ (C=8) | $\begin{bmatrix} 256, 1{\times}1, 32 \\ 32, 3{\times}3, 32 \\ 32, 1{\times}1, 256 \end{bmatrix}$ (C=8), ×2 |
| Average Pool
100 fc, softmax | Average Pool
100 fc, softmax |

Also for these experiments, we follow the data augmentation strategy described in (Xie et al., 2017). The input image has size 224x224 and it is randomly cropped from the resized original image. We use a mini-batch size of 256 on 8 GPUs (32 per GPU), with a weight decay of 0.0001 and a momentum of 0.9. We use *four* incremental training phases with a total of 120 epochs. In *phase 1* we train the model for 30 epochs with a learning rate of 0.1 for the convolutional and fully-connected layers, and a learning rate of 0.2 for the masks. In *phase 2* we finetune the model from *phase 1* for another 30 epochs with a learning rate of 0.1 and a learning rate of 0.0 for the masks (i.e., we use the fixed connectivity learned in phase 1). In *phase 3* we finetune the weights from *phase 2* for 30 epochs with a learning rate of 0.01 and the learning rate of the masks is 0.0. Finally, in *phase 4* we finetune the weights from *phase 3* for 30 epochs with a learning rate of 0.001 while the learning rate of the masks is still set to 0.0.

### E.3 ARCHITECTURES AND SETTINGS FOR EXPERIMENTS ON MINI-IMAGENET

For the experiments on the Mini-ImageNet dataset, a 64x64 crop is randomly sampled from the scaled 84x84 image or its horizontal flip, with per-pixel mean subtracted (Krizhevsky et al., 2012). For testing, we use the center 64x64 crop. The specifications of the models are identical to the CIFAR-100 models used in the previous subsection, except that the first input convolutional layer in the network is followed by a max pooling layer. The models are trained on 8 GPUs with a mini-batch size of 256 (32 per GPU), with a weight decay of 0.0005 and momentum of 0.9. Similar to training CIFAR-100 dataset, we also adopt *four* incremental training phases with a total of 320 epochs.

