# OpenReview forum: "Connectivity Learning in Multi-Branch Networks"
_ICLR.cc/2018/Conference — Reject_

### Official Review · AnonReviewer1 · 2017-11-22
**Extension of ResNeXt but at a price of more complicated training protocols**

**Rating:** 5
**Confidence:** 5

**Review:**

The authors extend the ResNeXt architecture. They substitute the simple add operation with a selection operation for each input in the residual module. The selection of the inputs happens through gate weights, which are sampled at train time. At test time, the gates with the highest values are kept on, while the other ones are shut. The authors fix the number of the allowed gates to K out of C possible inputs (C is the multi-branch factor in the ResNeXt modules). They show results on CIFAR-100 and ImageNet (as well as mini ImageNet). They ablate the choice of K, the binary nature of the gate weights.

Pros:
(+) The paper is well written and the method is well explained
(+) The authors ablate and experiment on large scale datasets

Cons:
(-) The proposed method is a simple extension of ResNeXt
(-) The gains are reasonable, yet not SOTA, and come at a price of more complex training protocols (see below)
(-) Generalization to other tasks not shown

The authors do a great job walking us through the formulation and intutition of their proposed approach. They describe their training procedure and their sampling approach for the gate weights. However, the training protocol gets complicated with the introduction of gate weights. In order to train the gate weights along with the network parameters, the authors need to train the parameters jointly followed by the training of only the network parameters while keeping the gates frozen. This makes training of such networks cumbersome.

In addition, the authors report a loss in performance when the gates are not discretized to {0,1}. This means that a liner combination with the real-valued learned gate parameters is suboptimal. Could this be a result of suboptimal, possibly compromised training?

While the CIFAR-100 results look promising, the ImageNet-1k results are less impressive. The gains from introducing gate weights in the input of the residual modules vanish when increasing the network size.

Last, the impact of ResNeXt/ResNet lies in their ability to generalize to other tasks. Have the authors experimented with other tasks, e.g. object detection, to verify that their approach leads to better performance in a more diverse set of problems?

---

> ### Author Response · Authors · 2018-01-05
> **Reply to AnonReviewer1**
>
> We thank the reviewer for the useful observations. We address the individual questions/comments below.
>
> * “The proposed method is a simple extension of ResNeXt.”
>
> We point out that our algorithm is a general procedure for connectivity learning in multi-branch networks. We chose to demonstrate it on ResNext, since it is one of the state-of-the-art architectures for image categorization. However, the method can be applied without modifications to any other multi-branch architecture. Furthermore, it is not even tied to the problem of image categorization, as it can use any arbitrary loss function. Thus, we disagree with the characterization that it is merely an extension of ResNeXt.
>
> * “Loss in performance when the gates are not discretized.”
>
> The training algorithm for the binary gates (GateConnect as shown in Algorithm 1) is substantially different from the training algorithm used to train the model with real-valued gates (mentioned in page 6). First, GateConnect performs sampling of gate weights to activate K branches while the training with real-valued gates does not use sampling since all branches are activated at all times. Second, in GateConnect the gradient of the loss function is calculated w.r.t. to the binary gate values (as shown in Algorithm 1, parameter update step); whereas in the case of training with real-valued gates, the gradient of the loss function is calculated w.r.t. the real-valued gates. Therefore, the loss in performance is not due to suboptimality. It is due to the different training procedure. We found that the forward and backward propagation using stochastically-sampled binary gates yields a larger exploration of connectivities and results in bigger changes of the auxiliary real-valued gates, which in turn leads to better connectivity learning.
>
>
> * “The training protocol gets complicated with the introduction of gate weights... The authors need to train the parameters jointly followed by the training of only the network parameters while keeping the gates frozen. This makes training of such networks cumbersome.”
>
> Overall, the proposed procedure remains straightforward. Evidence of this is the fact the entire algorithm used in the first stage can be summarized in a few lines of pseudo-code, as illustrated in Algorithm 1. The second stage simply involves freezing the binary gate weights and performing standard backpropagation. Considering that this two-stage procedure results consistently in a gain in accuracy, we believe that it will be of interest to the community despite the slightly increase in complexity. We note that we will be releasing the software of our approach for reproducibility and to allow other researchers to use it without having to reimplement it.
>
> * “The impact of ResNeXt/ResNet lies in their ability to generalize to other tasks. Have the authors experimented with other tasks, e.g. object detection?“
>
> In order to show the generalization ability of our approach, in this work we conducted experiments using many different model specifications and a wide variety of datasets, albeit all focused on image categorization. We plan to apply our approach to other tasks in future work.

---

### Official Review · AnonReviewer2 · 2017-12-02
**Interesting paper, not really gating, not justified.**

**Rating:** 5
**Confidence:** 4

**Review:**

The paper is clear and well written.
It is an incremental modification of prior work (ResNeXt) that performs better on several experiments selected by the author; comparisons are only included relative to ResNeXt.

This paper is not about gating (c.f., gates in LSTMs, mixture of experts, etc) but rather about masking or perhaps a kind of block sparsity, as the "gates" of the paper do not depend upon the input: they are just fixed masking matrices (see eq (2)).

The main contribution appears to be the optimisation procedure for the binary masking tensor g. But this procedure is not justified: does each step minimise the loss? This seems unlikely due to the sampling. Can the authors show that the procedure will always converge? It would be good to contrast this with other attempts to learn discrete random variables (for example, The Concrete Distribution: Continuous Relaxation of Continuous Random Variables, Maddison et al, ICLR 2017).

---

> ### Author Response · Authors · 2018-01-05
> **Reply to AnonReviewer2**
>
> We thank you for the insightful comments.
>
> * “Comparisons are only included relative to ResNeXt.”
>
> Since in the paper we chose to apply our connectivity learning to ResNeXt architectures, we use the ResNeXt performance as a baseline to assess the accuracy gain enabled by our method.
>
>
> * “This paper is not about gating but rather about masking... ”
>
> This is a good point. We changed “gate” to “mask” in the updated version of our paper.
>
>
> * “The main contribution appears to be the optimisation procedure for the binary masking tensor g. But this procedure is not justified: does each step minimise the loss? This seems unlikely due to the sampling. Can the authors show that the procedure will always converge? It would be good to contrast this with other attempts to learn discrete random variables (for example, The Concrete Distribution: Continuous Relaxation of Continuous Random Variables, Maddison et al, ICLR 2017).”
>
> The main contribution of our work is not a method to learn discrete random variables, but rather an algorithm for connectivity learning. To achieve this goal we do make use of learnable discrete random variable. It could very well be that other discrete optimization methods will lead to further improvements in our connectivity learning framework. But testing such methods is beyond the scope of this work, which is merely focused on the application of connectivity learning rather than optimization of discrete random variables.
>
> * “Does each step minimise the loss? This seems unlikely due to the sampling.”
>
> The algorithm is not guaranteed to reduce the loss at each iteration. But the deep learning literature includes many examples of methods/procedures that have no guarantee of reducing the original loss and yet are routinely adopted in practice due to their empirical effectiveness. Examples include dropout or batch normalization. Similarly, we believe that our method may be a useful tool in certain scenarios, given that it enables consistent accuracy improvements at a small additional computational cost.

---

### Official Review · AnonReviewer3 · 2017-12-04
**Another image classification architecture**

**Rating:** 5
**Confidence:** 4

**Review:**

The paper proposes replacing each layer in a standard (residual) convnet with a set of convolutional modules which are run in parallel.  The input to each model is a sparse sum of the outputs of modules in the previous set.  The paper shows marginal improvements on image classification datasets (2% on CIFAR, .2% on ImageNet) over the ResNeXt architecture that they build on.

Pros:
- The connectivity is constrained to be sparse between modules, and it is somewhat interesting that this connectivity can be learned with algorithms similar to those previously proposed to learn binary weights.  Furthermore, this learning extends to large-scale image datasets.
- There is indeed a boost in classification performance, and the approach shows promise for automatically reducing the number of parameters in the network.

Cons:
- Overall, the approach seems to be an incremental improvement over the previous work ResNeXt.
- The datasets used are not very interesting: Cifar is too small, and ImageNet is essentially solved.  From the standpoint of the computer vision community, increasing performance on these datasets is no longer a meaningful objective.
- The modifications add complexity.

The paper is well written and conceptually simple.  However, I feel the paper demonstrates neither enough novelty nor enough of a performance gain for me to advocate acceptance.

---

> ### Author Response · Authors · 2018-01-05
> **Reply to AnonReviewer3**
>
> We thank the reviewer. We address the questions/comments below.
>
> * “The approach seems to be an incremental improvement over the previous work ResNeXt...  I feel the paper demonstrates neither enough novelty nor enough of a performance gain for me to advocate acceptance.”
>
> Our approach is not an incremental improvement over ResNeXt. It is a general procedure to learn connectivity in multi-branch architectures. We chose to demonstrate it using ResNeXt architectures due to their strong performance. But we expect our method to be beneficial for other multi-branch models. Furthermore we note that accuracy gains are consistently obtained in all our experiments. Thus, we believe that researchers would be interested in using our method where even moderate performance improvements are critical. Finally, we are unaware of any other connectivity learning algorithm using an approach closely similar to ours. Thus, we disagree with the criticism of scarce novelty.
>
>
> * ”The datasets used are not very interesting: Cifar is too small, and ImageNet is essentially solved.  From the standpoint of the computer vision community, increasing performance on these datasets is no longer a meaningful objective.”
>
> We believe that few computer vision researchers would agree with this statement. While deep networks have achieved super-human performance on ImageNet, object categorization is far from being considered a solved problem and ImageNet remains today the most established benchmark for this task. Finally, we want to point out that our approach is very general and it is applicable without modifications to other tasks, different from image categorization. We chose to validate it on image categorization merely because of our interest in this application area and because manual design of CNNs for image analysis remains today a challenging endeavor.

---

### Decision · Program_Chairs · 2018-01-29
**ICLR 2018 Conference Acceptance Decision**

**Decision:**

Reject

**Comment:**

The paper proposes a method for learning connectivity in neural networks, evaluated on the ResNeXt architecture. The novelty of the method is rather limited, and even though the method has been shown to improve on the ResNeXt baselines on CIFAR-100 and ImageNet classification tasks (which is encouraging), it should have been evaluated on more architectures and datasets to confirm its generality.